# The Era of DAAs: Assessing the Patients’ Characteristics, Clinical Impact, and Emergence of Comorbidities in HIV/HCV-Coinfected versus HIV-Infected Individuals

**DOI:** 10.3390/jcm13133936

**Published:** 2024-07-04

**Authors:** Beatriz Álvarez-Álvarez, Laura Prieto-Pérez, Alberto de la Cuadra-Grande, Miguel Ángel Casado, Alfonso Cabello Úbeda, Aws W. Al-Hayani, Irene Carrillo Acosta, Ignacio Mahillo-Fernández, Miguel Górgolas Hernández-Mora, Jose M. Benito, Norma Rallón

**Affiliations:** 1Division of Infectious Diseases, Hospital Universitario Fundación Jiménez Díaz, 28040 Madrid, Spainmgorgolas@fjd.es (M.G.H.-M.); 2Pharmacoeconomics & Outcomes Research Iberia (PORIB), Paseo Joaquín Rodrigo 4, Letter I, Pozuelo de Alarcón, 28224 Madrid, Spain; adelacuadra@porib.com (A.d.l.C.-G.);; 3Biostatistics and Epidemiology Unit, Instituto de Investigación Sanitaria, Hospital Universitario Fundación Jiménez Díaz, Universidad Autónoma de Madrid (IIS-FJD, UAM), 28040 Madrid, Spain; 4HIV and Viral Hepatitis Research Laboratory, Instituto de Investigación Sanitaria Fundación Jiménez Díaz, Universidad Autónoma de Madrid (IIS-FJD, UAM), 28040 Madrid, Spain; 5Hospital Universitario Rey Juan Carlos, 28933 Móstoles, Spain

**Keywords:** comorbidities, DAA, direct-acting antivirals, HIV/HCV coinfection, mortality, non-AIDS-related events

## Abstract

**Objective:** To determine whether HIV-infected individuals versus individuals with HIV/HCV coinfection, in the era of interferon-free therapies, exhibit an increased incidence of comorbidities and non-AIDS-related events. **Methods:** A retrospective analysis was conducted by collecting data from clinical records of Spanish patients at a tertiary hospital involving HIV/HCV-coinfected and HIV-infected patients, all with effectively controlled HIV. Coinfected patients underwent HCV clearance using direct-acting antivirals (DAAs) and had no history of interferon treatment. The incidences of hypertension, diabetes mellitus, cardiovascular disease, kidney disease, liver disease, non-AIDS cancer, and death were compared between the groups. Multivariate adjustments for all factors potentially impacting outcomes were used to assess the risk of clinical event onset. Propensity score (PS) analyses were also conducted to support the multivariate model results. **Results:** Data were available from 229 HIV/HCV-coinfected patients and 229 HIV-infected patients. Both cohorts were comparable in terms of age, gender distribution, follow-up, and HIV-related characteristics. Multivariate models and PS showed that previous exposure to HCV was not associated with the onset of any clinical events studied. Significant differences between HIV/HCV-coinfected and HIV-infected were not found for survival according to the log-rank test (*p* = 0.402). **Conclusions:** Successful HCV elimination using DAAs improved the outlook regarding comorbidities and survival across HIV/HCV-coinfected cohorts. Early HCV detection and DAA therapy could enhance clinical results. These findings provide an optimistic perspective for those living with HIV/HCV coinfection and underscore the importance of continuing efforts toward early detection and DAA treatment initiation.

## 1. Introduction

Over the course of the last four decades of the human immunodeficiency virus (HIV) pandemic, various milestones have been achieved that have improved the disease prognosis, resulting in an estimated 39 million people living with the infection worldwide [1].

Since the introduction of effective antiretroviral treatment (ART), HIV infection has become a chronic disease, and the longevity and quality of life of people living with HIV (PLWH) have increased significantly. However, ART is still unable to completely restore life expectancy in infected individuals compared with non-infected individuals, mainly due to the development of comorbidities [2]. The comorbidity-free life expectancy is markedly reduced in PLWH [3], and comorbidities or nonacquired immune deficiency syndrome (AIDS) related events appear earlier and more frequently than in the general population [4,5]. Among the comorbidities that were found to be determinants of death in PLWH [6,7,8], hepatitis C virus (HCV) coinfection is an important risk factor [9,10,11].

HCV infection is especially prevalent among PLWH, as both HIV and HCV share the same routes of transmission. Approximately 2.9 million PLWH worldwide are estimated to be coinfected with hepatitis C [12]. While HCV infection mostly affects people who inject drugs (PWID), an increase in sexually transmitted hepatitis C outbreaks has occurred during the past two decades, largely among HIV-infected men who have sex with men (MSM) [13]. This increase in new cases of sexual transmission, especially among HIV-infected MSM, is now considered an epidemic in Europe, Australia, North America, and Asia. Many of these cases are linked to high-risk sexual behaviors such as condomless anal intercourse with multiple partners, facilitated by the use of mobile geo-social networking applications; the ChemSex practice, involving the concurrent use of stimulant drugs such as GHB or mephedrone for sexual purposes; or an increase in injecting drug use in these settings (“SlamSex”) [14,15,16,17,18]. This phenomenon poses an additional public health problem as it represents an active source of new hepatitis C cases, especially in industrialized countries. Furthermore, these practices are the main drivers of HCV reinfections [4].

On the other hand, HIV infection induces a state of persistent chronic immune activation due to irreversible damage at the onset of infection that persists regardless of the efficacy of the ART regimen used. Among other causes, high levels of other pathogens coinfecting the patient, such as HCV, are the main drivers of immune system distress. This inflammatory environment causes functional damage to other organs through several potential pathways and is a key factor in the development of comorbidities and clinical events unrelated to AIDS [19,20,21].

In addition to causing additional stress on the immune system, HIV/HCV coinfection establishes interactions between viruses that can modify the clinical course of both diseases [22]. The impact of HIV on hepatitis C has been addressed in several studies, showing an increase in the HCV viral load (VL), increased viral persistence, and accelerated development of liver fibrosis, cirrhosis, and hepatocellular carcinoma, leading to increased HCV-related mortality [23,24,25]. Similarly, HCV can alter the clinical progression of HIV infection [26] and the incidence of some non-AIDS comorbidities [27]. This has been linked to the effect of HCV coinfection on different parameters of HIV pathogenesis, which manifests as increased levels of T-cell activation and exhaustion [28,29,30]. Coinfection with HCV has also been shown to have a negative impact on biomarkers of systemic inflammation, coagulopathy, and endothelial activation in HIV/HCV-coinfected patients [31]. Additionally, although more controversial, there might be an effect of HCV coinfection on HIV [32,33,34,35]. This complex situation could lead to dysregulation of the immune response, accelerating the progression of HIV infection and, therefore, favoring the development of non-AIDS-related comorbidities [27].

To date, most studies on the impact of HIV/HCV coinfection have included patients receiving traditional therapy comprising interferon-α (IFN-α) with suboptimal sustained virological response (SVR) rates. IFN-α is a protein with proinflammatory and immunomodulatory activity [36], which may introduce bias when evaluating the impact of HCV eradication on the immune-virological profile and prognosis of coinfected patients. Since 2014–2015, therapeutics for hepatitis C have changed radically with the incorporation of direct-acting antivirals (DAAs). These new therapeutic regimens have enabled the eradication of HCV in a rapid, highly effective, and IFN-free manner. Thus, there is an opportunity to study the real effect of HCV eradication on the natural history of HIV infection in patients coinfected with both viruses, eliminating the potential bias of the immunomodulatory effect of IFN-α. Within this context, the aim of the present study was to determine the relationship between HCV eradication with DAAs and the frequency of non-AIDS comorbidities and mortality in PLWH.

## 2. Materials and Methods

### 2.1. Study Design and Participants

This was a retrospective, observational study carried out by collecting data from the clinical history of patients who attended a tertiary hospital in downtown Madrid (Spain), which provides care for more than 3500 people with HIV. Those HIV/HCV-coinfected patients were compared to individuals infected exclusively by HIV. Both coinfected and monoinfected cohorts had similar mean age, gender distribution, mean follow-up time, and clinical characteristics related to HIV infection. All patients were receiving ART and exhibited good control of HIV infection during the follow-up. The data were collected from the clinical records from January 2023 to September 2023. Thus, no additional measures to those compiled in clinical practice were gathered.

Inclusion criteria for HIV-infected patients only included a previous diagnosis of the infection, no history of HCV infection, and current control of the disease using ART. Regarding the HIV/HCV group, eligibility criteria included adult PLWH diagnosed with HIV/HCV coinfection, in whom HCV was eradicated by any DAA. Patients who had previously received IFN-α were excluded. Complications at baseline were not considered as exclusion criteria in order to adjust statistical analysis to the previous presence of those comorbidities. Regarding the HIV infection in those HIV/HCV-coinfected patients, adequate control of the disease with ART was also considered an inclusion criterion.

The study was approved by the Hospital Ethics Committee (EO 228-23_FJD) on 12 September 2023. The research was conducted in accordance with the General Data Protection Regulation (GDPR) and the Declaration of Helsinki guidelines regarding personal data protection.

### 2.2. Measures

**Demographics and lifestyle habits**—Data on participants’ age, gender, ethnicity, and sexual orientation were collected. Similarly, data on patients’ lifestyle habits, including smoking, alcohol consumption, and drug use, were obtained from their clinical history. Information regarding the purposes of parenteral drug use (nonconsumer, conventional use, or ChemSex) and the specifics of drug consumption were also gathered.**HIV infection**—HIV infection was characterized by the date of diagnosis, route of transmission, HIV clinical stage, CD4+ cell count at ART initiation (cells/µL), CD4+/CD8+ ratio at ART initiation, baseline HIV-1 VL (copies/mL), and history of AIDS. The presence of AIDS-related illnesses (tuberculosis, recurrent pneumonia, *Pneumocystis* pneumonia, esophageal candidiasis, Kaposi’s sarcoma, cryptococcosis, cerebral toxoplasmosis, non-Hodgkin lymphoma and progressive multifocal leuco-encephalopathy) was also recorded.**HIV treatment**—The date when the patient started ART, the patient’s time spent on ART, and the time elapsed from HIV diagnosis to the initiation of ART were recorded. Given that previous treatment with rilpivirine-based regimens is associated with clinical benefit for improving liver stiffness (LS) [37], data were also gathered on whether the patient had ever been exposed to rilpivirine or not.**HVC infection**—HCV infection was characterized by the route of transmission (sexual, parenteral), date of diagnosis, stage of hepatitis C at diagnosis (acute or chronic), HCV genotype, HCV VL at DAA initiation (IU/mL) and range (<800,000 IU/mL, ≥800,000 IU/mL), antiviral regimen used (DAAs), time from diagnosis to treatment, and achievement of SVR at 12 weeks (SVR12). Additionally, the number of subsequent HCV reinfections was recorded.**Biochemistry and LS assessment**—The patients’ biochemical data included liver chemistry (aspartate aminotransferase [AST], alanine aminotransferase [ALT], gamma glutamyl transpeptidase [GGT]), renal function (glomerular filtration rate [GFR]), lipid profile (triglycerides, low-density lipoprotein [LDL], high-density lipoprotein [HDL] and total cholesterol levels), and complete blood count data, as well as HIV status (quantitative HIV RNA data, and CD4+ and CD8+ T-cell counts). The APRI score and fibrosis stage (no fibrosis: APRI < 0.5; moderate fibrosis: APRI 0.5–1.5; cirrhosis: APRI > 1.5) and the FIB-4 score and fibrosis stage (no fibrosis: FIB-4 < 1.45; moderate fibrosis: FIB-4 1.45–3.25; cirrhosis: FIB-4 > 3.25) were calculated as surrogate markers of liver disease. Biochemistry test results and fibrosis measurements (APRI and FIB-4) were collected for the HIV/HCV cohort before starting DAA therapy and after SVR12. Additionally, in HIV/HCV-coinfected patients, LS was assessed using transient elastography (FibroScan), the gold standard technique, before DAA therapy. Fibrosis classification was determined using the METAVIR scale, where the LS cutoff values were LS < 7.1 KPa for stage F0–F1, LS between 7.1 and 9.4 KPa for stage F2, LS between 9.5 and 12.4 KPa for stage F3, and LS ≥ 12.5 KPa for stage F4 of cirrhosis. Since FibroScan is not used as a routine test in HIV-monoinfected patients, alternative biochemical markers (APRI and FIB-4) were used to compare the groups [38].**Patients’ baseline comorbidities**—The data on HIV infection risk factors included the number and type of previous comorbidities at the time of diagnosis, such as hypertension, diabetes mellitus (DM), dyslipidemia, cardiovascular disease (CVD), kidney disease, liver disease, and non-AIDS cancer. Additionally, the number of documented sexually transmitted infections (STIs), serological data on hepatitis A and B, and obesity (determined by a body mass index [BMI] over 30 Kg/m^2^) were recorded.

### 2.3. Outcomes

**Development of comorbidities**—The clinical events registered in the medical history and gathered for this study were hypertension, DM, dyslipidemia, CVD, kidney disease, liver disease, non-AIDS cancer, and death. The number of clinical events that occurred during the follow-up and the overall follow-up time were also recorded. In HIV/HCV-coinfected patients, the comorbidities recorded were developed from HCV diagnosis to the present. In HIV-monoinfected patients, the comorbidities presented at baseline were not considered.

### 2.4. Statistical Analyses

The differences between the HIV and HIV/HCV groups were expressed as the mean or median and accompanied by the standard deviation (SD) or interquartile range (IQR), respectively. Inference tests were conducted to measure the differences between groups regarding age, gender distribution, follow-up time, and HIV-related variables. It was expected that no significant differences between groups would be found since data collection from the clinical records was based on comparable patients for these characteristics. Chi-squared or Fisher’s exact test was also used to assess the differences in categorical variables. For quantitative variables, before performing any inference analysis, the Shapiro–Wilk test was conducted to determine whether the data adjusted to a normal distribution. The nonparametric Mann–Whitney U test and parametric Student’s *t* test were performed to evaluate the differences between groups in nonnormally or normally distributed quantitative variables, respectively.

Given that multiple clinical and demographic features related to HCV infection are expected to be associated with comorbidity onset and death, multivariate logistic regression models were designed to quantify the relationship between each comorbidity and the other variables. In those analyses, the variable group (HIV vs. HIV/HCV) was included in all logistic models. For the remaining variables, potential risk and protective factors were considered, and subsequently, a stepwise method was used to determine the relevant variables to be included in the models. To measure the frequency of clinical events in both the HIV and HIV/HCV groups, odds ratios (ORs) were estimated, accompanied by their respective confidence intervals (CIs). Patients diagnosed with the comorbidity of interest at baseline were excluded from that specific comorbidity model development.

In order to measure risk and protective factors associated with clinical events, additional subgroup-specific multivariate logistic models were developed for the HIV and HIV/HCV groups. Although Cox models should be used for time-to-event variables, in both cases, logistic models were developed because of the lack of time-to-event data on the patient’s clinical records regarding the studied comorbidities.

Additional analyses included propensity score (PS) techniques to balance HIV and HIV/HCV cohorts. The PS was estimated using logistic regression models, which included the group (monoinfected vs. coinfected) as the dependent variable. The remaining factors were considered to be independent variables in the models; these factors were significantly different between groups and were associated with the dependent variables (group: HIV vs. HIV/HCV) [39].

Patients’ survival was also assessed using the Kaplan—Meier method to obtain time-event curves. The survival distributions were compared between the study groups by performing a log-rank test.

One-sided analyses were performed for inference tests considering an alpha of 0.05 to determine statistical significance. The analyses were performed in R (V.4.2.2).

## 3. Results

Data were collected from 458 PWLH on ART; 229 patients composed the HIV group, and 229 were included in the HIV/HCV group. The demographic and baseline clinical characteristics of the patients in the HIV and HIV/HCV groups are presented in Table 1.

As expected, no significant differences were found for age (*p* = 0.627) and gender (*p* = 1.000). Moreover, there were no differences between the two groups regarding the course of HIV infection, covering aspects such as time since HIV diagnosis (*p* = 0.092), clinical stage (*p* = 0.051); CD4+ T-cell count (*p* = 0.845), CD4+/CD8+ ratio (*p* = 0.313), and HIV-1 VL at ART initiation (*p* = 0.102); time since HIV diagnosis to ART initiation (*p* = 0.155); time at HIV treatment (*p* = 0.573); ever exposure to a rilpivirine-containing regimen (*p* = 0.287); and preexisting comorbidities (including hepatitis A, hepatitis B, obesity, dyslipidemia, and CVD). In contrast, differences were found for the HIV route of transmission (*p* < 0.001), CD4+ cell count at ART initiation (*p* = 0.012), history of AIDS (*p* = 0.025), and poor previous adherence to ART (*p* = 0.001).

The sociodemographic characteristics of the two groups were largely similar; however, the coinfected group had a higher rate of heterosexual transmission and a significantly higher proportion of patients who ever used drugs (74.1% of HIV/HCV-coinfected individuals, *p* < 0.001), including substances such as cocaine, mephedrone, GHB/GBL, and popper; these drugs were often administered parenterally, with 22.6% reporting conventional use (mainly cocaine and heroin) and 22.2% reporting “ChemSex” practices (“SlamSex”). Smoking and alcohol consumption were also more common in the coinfected group. Additional demographic features concerning AIDS conditions and specific drug use are presented in Appendix A, respectively.

The episodes of HCV infection were diagnosed between 1985 and 2022, and their clinical characteristics are described in Table 1. The infection was sexually transmitted in 77.7% of the individuals (178/229), and all of them were MSM. Hepatitis C was diagnosed in the acute phase of the infection in 65.5% of patients (150/229). In this setting, 13.1% of patients presented HCV reinfections, and 2.2% of them suffered three events. The mean time since HCV diagnosis and successful DAA therapy was 5.66 years (SD = 9.13), and the maximum reported time was 35 years. The first-line DAA therapy was effective in 98.2% of patients; the remaining patients achieved HCV clearance after a second-line treatment with DAAs.

At the beginning of the anti-HCV treatment with DAAs, 61.6% of patients had a VL greater than 800,000 IU/mL, and 29.7% had an LS greater than 7.1 Kpa, as measured using FibroScan; these patients had fibrosis stages F2 to F4 according to the METAVIR scale. Advanced fibrosis was present in 4.8% of patients.

Regarding baseline comorbidities, significant differences between the study groups were found for the number of diagnosed comorbidities at baseline (*p* < 0.001), number of documented STIs (*p* < 0.001), and presence of hypertension (*p* = 0.002), DM (*p* = 0.040), kidney disease (*p* = 0.040), liver disease (*p* < 0.001), and non-AIDS cancer (*p* = 0.007) at baseline. The HIV/HCV-coinfected patients had a higher number of comorbidities, documented STIs and a greater frequency of those comorbidities described for which significant differences were observed.

During the follow-up, all coinfected patients maintained an undetectable HIV-1 VL (VL < 50 copies/mL), except for seven who experienced viral blips with VL > 200 copies/mL and one patient whose initial HIV-1 VL was 1530 copies/mL at the start of DAA therapy, which later became undetectable. After DAA therapy and achieving SVR12, four coinfected patients exhibited viral blips with HIV-1 VL < 200 copies/mL, and three different patients had VLs > 200 copies/mL (419, 6565, and 13,600 copies/mL). All these patients continued their clinical follow-up, which recovered successful immunovirological control. For the monoinfected group, all patients had a VL < 50 copies/mL, which was an inclusion criterion.

The clinical outcomes of patients in the HIV/HCV cohort after achieving SVR12 with DAA therapy are presented in Table 2. Several differences in outcome achievement were observed between the HIV/HCV and HIV groups. Compared with coinfected patients, monoinfected patients exhibited higher ALT levels (*p* < 0.001), total (*p* = 0.010), LDL (*p* = 0.007) and HDL-cholesterol concentrations (*p* = 0.015), and CD4+/CD8+ ratios (*p* < 0.001). Compared with the monoinfected group, the HIV/HCV group showed higher GFR (*p* < 0.001), platelet (*p* = 0.011), and CD8+ cell counts (*p* = 0.001). Additionally, the coinfected patients presented higher hepatic fibrosis severity measured by the APRI (*p* < 0.001) and FIB-4 scores (*p* < 0.001). No other statistically significant differences were found for clinical and biochemical features after SVR12.

The cohorts were followed to study the development of comorbidities over a median of 14 vs. 13 years in coinfected and monoinfected groups, respectively. No differences were found for the follow-up period (HIV/HCV vs. HIV group, median [IQR]: 14.0 [8.0–19.0] vs. 13.0 [8.0–18.0]; *p* = 0.230). When logistic models were adjusted for other significant variables, neither HIV/HCV coinfection nor HIV monoinfection was significantly associated with the development of any comorbidity. However, other variables were significantly associated with the development of comorbidities (Table 3).

The results of the subgroup multivariate logistic regressions, which included the variables significantly associated with the development of each comorbidity in both groups, are shown in Table 4.

After obtaining the balance between the HIV and HIV/HCV cohorts through PS analysis, the results revealed no statistically significant associations between the group (HIV vs. HIV/HCV) and the development of the comorbidities studied (Figure 1).

A total of 10 HIV/HCV-coinfected patients (4.4%) and three HIV-monoinfected patients (1.3%) died during follow-up. The Kaplan–Meier estimates are depicted in Figure 2. No significant differences were found for survival according to the log-rank test (*p* = 0.402). The adjusted OR was 0.25 (95% CI = 0.04–1.49; *p* = 0.128) in the multivariate analysis (Table 3), but statistical significance was not found for the relationship between the group (HIV vs. HIV/HCV) and death. The PS analysis confirmed the lack of association between these variables (Figure 2). The number of documented STIs was the only variable that was significantly related to death (OR = 0.41; 95% CI = 0.21–0.82; *p* = 0.011) and was also the only variable that was significantly related to death in the HIV/HCV analysis (OR = 0.39; 95% CI = 0.18–0.84; *p* = 0.017). Additional variables were found to be related to death in subgroup multivariate logistic models (Table 4).

## 4. Discussion

This retrospective cohort study provides insight into the clinical impact of the clearance of HCV via DAA therapy in PLWH receiving ART. To the knowledge of the authors, this research is the first to investigate the impact of HCV among patients with chronic HIV infection on the risk of developing comorbidities and mortality after SVR12 was achieved exclusively via DAA therapy. The main findings were as follows: (1) several differences between HIV-monoinfected and HIV/HCV-coinfected patients exist at baseline, which helps to characterize and understand the population coinfected with HIV/HCV; (2) after hepatitis C treatment with DAAs achieving SVR12, although patients might recover from their liver disturbances and exhibit improved CD4 counts, inflammation, immune activation, and some liver fibrosis persist in the short-medium term; and (3) clinical, biochemical, and immunological differences at baseline and after SVR12 are associated with a greater risk of liver disease onset and death in HIV/HCV-coinfected patients, but previous exposure to HCV in PLWH is not associated with the development of any clinical event studied.

The present study aimed to analyze the influence of HCV on the natural history of HIV and the effect of HCV eradication. Thus, a cohort of PLWH suffering from hepatitis C who cleared HCV with DAAs achieving SVR12 was followed, and the development of non-AIDS events and mortality were monitored. These coinfected patients were compared to a cohort of HIV-monoinfected patients and matched by age, gender, and follow-up time. 

### 4.1. Patients’ Clinical Characteristics and Comorbidities at Baseline

According to the HIV baseline characteristics of the patients, the mean CD4+ T-cell count and CD4+/CD8+ ratio at ART initiation were less than 500 cells/µL and 0.4, respectively. Similar results were found in both groups, except for a significantly higher proportion of coinfected patients exhibiting CD4+ T-cell counts less than 200 cells/µL, which could explain the increase in AIDS-related events recorded in this group. After starting ART, both groups achieved successful immune recovery, as measured by a CD4+ T-cell count greater than 500 cells/µL, highlighting the effectiveness of the treatment even when poorer adherence to ART is expected in coinfected patients. Moreover, the parental route of transmission was more frequent in coinfected patients, which could be explained by the patient’s lifestyle habits and awareness of their health status, as well as the poor adherence of coinfected patients to ART.

Additionally, statistically and clinically significant differences at baseline were found in lifestyle habits, which are well-known risk factors for hepatitis C exposure and transmission. A higher proportion of smokers and alcohol and parenteral drug users were observed in the HIV/HCV group, where there was also a higher proportion of heterosexual patients exposed to HIV via the parenteral route of transmission due to drug consumption. This cohort also exhibited a high rate of sexual transmission of hepatitis C, indicative of the epidemiological shift in the prevalence of HCV in industrialized countries [13], suggesting that sexual transmission is currently one of the primary active sources of new hepatitis C infections [14,15,16,17,18]. In this HIV/HCV cohort, at least three-quarters of the individuals acquired HCV through sexual transmission, and all of them were MSM. Notably, there was a significantly greater consumption of psychoactive drugs for sexual purposes, with mephedrone being the most used substance. Approximately one-quarter of the patients were administered this drug parenterally (“SlamSex”). As might be expected, according to these data, more STIs were documented in coinfected patients than in monoinfected patients. Moreover, 13.1% of coinfected patients reported HCV reinfection after successful eradication of a primary infection with DAAs, a rate consistent with that of other published cohorts [4].

To assess the effect of exposure to HCV on the development of non-AIDS-related events and HIV prognosis, risk factor information, such as baseline characteristics in those previously diagnosed with HCV infection, was collected, regardless of whether the diagnosis was performed prior to or after HIV infection and before HIV diagnosis in the monoinfected group. Despite the low prevalence of comorbidities at baseline in both groups, disparities were found in favor of the coinfected group for hypertension, DM, renal disease, liver disease, and non-AIDS cancer. The lifestyle and/or chronic HIV infection of these individuals could explain why certain risk factors were more frequent in HIV/HCV patients.

Some of the clinical variables for which differences were observed for mono- and coinfected patients at baseline and after achieving SVR12 represent well-known risk factors for developing the comorbidities of interest and even death. For instance, previous studies found that higher levels of LDL-cholesterol increase the risk of hypertension [40,41,42], which might explain the results obtained in the present study. The worse lipidic profile could also be a determinant of the increased risk of developing dyslipidemia [43]. Along the same line, the deterioration of GFR in these patients, in relation to coinfected patients, could lead to a poor prognosis for renal disease [44,45]. On the contrary, in the case of HIV/HCV-coinfected patients, pronounced liver impairment, as well as alcohol consumption and abuse of illegal substances [46,47], can be expected to be the main drivers of the increased risk of liver disease. Likewise, the HIV infection [48] and the smoking habit in coinfected patients might be related to the development of cardiovascular comorbidities [49].

These findings allowed for the characterization of the study population (HIV/HCV-coinfected) and the identification of groups at higher risk of HCV infection, which will facilitate early suspicion and diagnosis.

Before DAA therapy, higher levels of liver enzymes (AST, ALT, and GGT), platelet count, and increased levels of hepatic fibrosis markers (assessed using the APRI and FIB-4 scores) were found in HIV/HCV-coinfected patients compared with the PLWH cohort of patients who had never had hepatitis C, suggesting greater liver damage in the HIV/HCV group. In the coinfected individuals, LS data measured with FibroScan and pre-DAA treatment were available. A third of these patients presented measurements greater than 7.1 Kpa, and the mean LS exceeded normal values. The increased stiffness data may reflect either fibrosis or necro-inflammatory activity in the liver, both of which are indicative of liver injury. However, according to clinical practice, LS data remeasured by transient elastography was not available after SVR12 with DAAs in most coinfected patients, nor is this a common practice for HIV-monoinfected patients. Thus, other markers, such as the APRI or FIB-4 scores, were considered in this study to compare the two groups [38]. 

After DAA therapy, the whole cohort of coinfected patients reached SVR12. Most of these patients normalized transaminase levels and reduced LS, as calculated using the APRI and FIB-4 indices. In line with previous research findings, these results suggest that DAA therapy may allow the recovery of liver injury following HCV clearance [50]. Nonetheless, despite most patients experiencing a reduction in LS within the normal range, the LS measurements remained significantly greater in the coinfected group (according to both the APRI and FIB-4), which is indicative of more severe hepatic involvement, at least at SVR12 post-DAA therapy. 

In previous research conducted by our group, systemic inflammation and endothelial activation were compared between subgroups of patients coinfected with HIV/HCV and those infected with HIV alone, which were all included in the cohort presented in this work. Independent associations between IP-10 (a systemic inflammation marker) and GGT levels and between VCAM-1 (an endothelial activation biomarker) and AST levels were found, revealing the link between hepatic necro-inflammatory activity and endothelial activation in systemic inflammation during HIV/HCV coinfection [51]. This hepatic necro-inflammatory activity could be explained by the activity IP-10, a chemokine that induces the recruitment of immune cells in the liver tissue, and VCAM-1, which promotes the adhesion and recruitment of leukocytes to the vessel wall. Thus, the increased levels of AST and GGT are indicative of hepatic damage [51]. In addition to its association with liver damage, the association of VCAM-1 with CVD in HIV-infected populations has been well established [52]. 

In this analysis, the levels of VCAM-1 and IP-10 remained elevated post-DAA treatment despite the clearance of HCV and normalization of transaminases, especially in patients with higher LS. This suggests the persistence of a proinflammatory and procoagulant state, at least in the short term, which could result in a potentially greater incidence of CVD and other comorbidities among HIV/HCV-coinfected patients than among HIV-monoinfected individuals [51]. 

An interesting finding in the present study was the remarkable increase in CD4+ T-cell counts following HCV eradication to levels comparable to those in HIV-monoinfected patients. This finding supports the negative impact of HCV on immune restoration in patients on ART [53]. However, after patients achieved SVR12, the CD8+ T-cell count remained significantly higher, and the CD4+/CD8+ T-cell ratio was lower in coinfected patients. This persistent increase in CD8+ T-cell counts may not only indicate that immune activation is not influenced by IFN-based regimens but also suggests that CD8+ T cells may become dysfunctional. Previous research has reported the significant negative impact of HCV infection on CD8+ T-cell exhaustion in coinfected PLWH [54]. Our group carried out another study comparing several immunological parameters of HIV among subgroups of HIV/HCV-coinfected and HIV-infected patients, all of which were included in the cohort presented in this work. After HCV eradication, both the activation and exhaustion of CD8+ T cells remained significantly greater in coinfected patients. These results supported the fact that HCV clearance via treatment with DAAs does not fully reverse T-cell homeostasis alterations, at least in the short term [30]. In addition, T-lymphocyte exhaustion could influence viral escape from the immune response to both infections [55]. Therefore, both the activation and exhaustion of CD8+ T cells are significant factors in the pathogenesis of HIV and contribute to the progression of this infection, which could have substantial clinical implications for coinfected patients [54]. Finally, it was reported that a persistent CD8+ T-cell count and, consequently, a low CD4+/CD8+ T-cell ratio in HIV-infected patients, even with CD4+ T-cell count greater than 500 cells/µL, are predictors of non-AIDS events and a greater risk of mortality [56], which was also observed in this cohort. 

The presence of more AIDS-related events at HIV diagnosis, additional risk factors, harmful lifestyle habits, a higher prevalence of coinfections such as STDs, and underlying signs of systemic inflammation, immune activation, and disruption of cellular homeostasis could predispose the coinfected group to exhibit a more advanced clinical presentation of HIV. This could be characterized by a more frequent or earlier onset of non-AIDS-related events or even increased mortality in this group compared with the HIV-monoinfected group.

### 4.2. Clinical Impact and Emergence of Comorbidities Onset

To determine whether the risk of comorbidities onset and death was due to HCV exposure, multivariate logistic regression models were used to adjust for the development of clinical events of interest according to the remaining significant variables for each group. There was no significant association between the group (HIV vs. HIV/HCV) and the development of any of the comorbidities. Moreover, no association between group and mortality was observed in multivariate models and Kaplan–Meier curves. The PS analysis confirmed the absence of a relationship between HCV exposure and the development of comorbidities and/or mortality. 

In contrast, the variables associated with the clinical events were the number of documented STIs, number of previous comorbidities, CD8+ T-cell count, drug consumption, time since HIV diagnosis to starting ART, FIB-4 fibrosis stage, and time since HCV diagnosis to treatment with DAAs. It was particularly striking that the number of documented STIs was a protective factor in the models. It was hypothesized that patients with a greater health status could maintain a more active sexual life, which could explain the results obtained. Nonetheless, further research should be conducted in this regard. In any case, the risk factors for each clinical event studied, identified in subgroup multivariate logistic regression, support the fact that the development of these events is not directly related to HCV exposure.

Regarding the severity of fibrosis pre-DAA in coinfected patients, as measured by FibroScan, the APRI and FIB-4 scores, no associations were found between the liver fibrosis stage and the development of any event included in the study, except for hepatic disease. It should be noted that the proportion of individuals exhibiting advanced fibrosis within the cohort was low. However, relationships were found with other aspects of HCV infection, for instance, associations between the number of hepatitis C episodes and an elevated risk of CVD, the diagnosis during the chronic HCV stage and an increased risk of non-AIDS cancers, and time from HCV diagnosis to DAA therapy and a greater risk of mortality. These findings underscore the clinical advantage of timely HCV diagnosis and prompt initiation of eradication therapy. Moreover, a higher viral load prior to DAA therapy initiation was linked to the onset of dyslipidemia.

Finally, it should be noted that age was found to be a risk factor associated with many comorbidities’ onset, including hypertension [57], DM [58], dyslipidemia [59], kidney disease [60], non-AIDS cancer [61], and death, which is consistent with previous findings. Along the same line, obesity was found to be a risk factor for DM and dyslipidemia, which is also consistent with previous research [62,63].

### 4.3. Limitations

For an adequate interpretation of the results, several limitations of the present study should be addressed. 

First, the follow-up period of the study extended to approximately 10–15 years; thus, the long-term risk of comorbidity development and death should be evaluated again in further studies. Second, although the sample included in the study was considered large enough to provide statistical evidence, the data were collected from a single hospital, and the extrapolation of the results to other settings should be carried out with caution. For example, the number of women represented in our cohort was low according to epidemiological data. Third, the percentage of patients with advanced fibrosis due to hepatitis C was low, which might underrepresent a particularly vulnerable group [26]. However, the availability of highly effective drugs against hepatitis C, in addition to the current efforts motivated by the World Health Organization toward the elimination of HCV, promotes HCV screening. It should be expected that the diagnosis of this infection will be performed at increasingly earlier stages of infection, as was the case for this study’s cohort. Finally, due to the lack of time-to-event data on the clinical histories of patients regarding the comorbidities, logistic models were developed for the analyses instead of using Cox proportional hazards models.

Despite these issues, the findings of the present study are considered robust because exposure to HCV does not have a meaningful impact on the risk of developing comorbidities, or even dying, in PLWH following HCV eradication with DAAs.

### 4.4. Strengths

The population included in the present study consisted of two homogenous cohorts of HIV-infected patients, who all received successful long-term ART and exhibited high and stable CD4+ T-cell counts. We provide prospective information on pathogenic or immunovirological aspects related to systemic inflammation, T-cell function, homeostasis, and reservoir size [30,51,64] for this cohort of both coinfected and monoinfected individuals. Scientific and clinically relevant findings about these infections were published previously, allowing us to establish the hypotheses that motivated the present study, thereby completing a comprehensive and thorough investigation of the impact of HCV in HIV/HCV coinfection.

The HIV/HCV cohort allows the drawing of current conclusions, as it is framed in the era of DAAs, which are currently positioned as first-line drugs and avoid the bias caused by the immunomodulatory effect of IFN-α. Furthermore, the profile of patients in the coinfected group is representative of new epidemiological trends within the context of HCV infection in PLWH, posing a significant challenge to the eradication of hepatitis C in industrialized countries. Therefore, the conclusions of this study are useful and practical for the future management of hepatitis C.

Given the availability of IFN-free therapies, the SVR12 rate observed for PLWH with HCV demonstrates the success of the treatment, which is also similar to the rate observed in patients without infection by HIV [65]. Clearing HCV not only confers a proven clinical benefit in PLWH [66] but also improves patient-reported outcomes [67], even in patients with significant liver impairment [68]. Moreover, the interaction between HIV treatments and DAAs is easily managed in most cases and does not interfere with HCV clearance [49]. Within this context, all the available evidence indicates that DAAs represent an opportunity to improve the health status and life expectancy levels of HIV/HCV-coinfected patients to those of HIV-monoinfected individuals.

## 5. Conclusions

Exposure to HCV in PLWH on ART does not have an impact on the development of hypertension, DM, dyslipidemia, CVD, kidney disease, liver disease, non-AIDS cancer, or mortality in the short-medium term when patients achieve SVR12 after DAA therapy. Consequently, exposure to HCV in PLWH does not accelerate the progression of HIV infection toward non-AIDS events, nor does it trigger a negative impact on survival if treated in a timely manner. In HIV/HCV-coinfected patients, the eradication of HCV could enhance the effectiveness of immune recovery, which follows the control of HIV replication. Although it is advisable to conduct studies extending the follow-up time to strengthen these findings, early diagnosis of HCV in PLWH and prompt DAA treatment to cure hepatitis C, along with the promotion of healthy lifestyle habits, are highly recommended. This approach is crucial not only for preventing liver damage but also for maintaining the health status and life expectancy of these patients at levels comparable to those of HIV-monoinfected patients. Moreover, this strategy could help curb ongoing HCV transmission in certain populations, contributing to HCV microelimination.

## Figures and Tables

**Figure 1 jcm-13-03936-f001:**
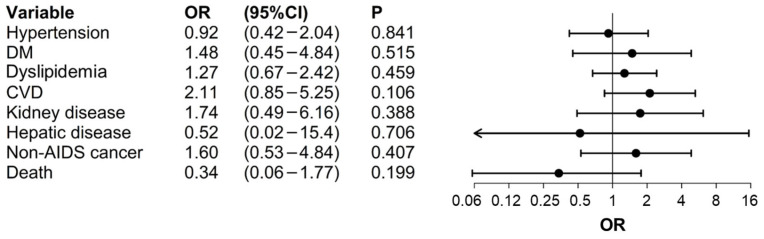
PS analysis results for the risk of comorbidities development and mortality: HIV vs. HIV/HCV cohorts. **Abbreviations.** AIDS: Acquired Immunodeficiency Syndrome; CI: Confidence Interval; CVD: Cardiovascular Disease; DM: Diabetes Mellitus; HCV: Hepatitis C Virus; HIV: Human Immunodeficiency Virus; OR: Odds Ratio; PS: Propensity Score. **Results interpretation**. OR < 1 represents a higher risk for the clinical event development in HIV/HCV-coinfected patients (lower risk for HIV-monoinfected patients); OR > 1 represents a lower risk for the clinical event development in HIV/HCV-coinfected patients (higher risk for HIV-monoinfected patients); 95% CI including 1 indicate no statistically significant association between group and clinical event onset (*p*-value > 0.05).

**Figure 2 jcm-13-03936-f002:**
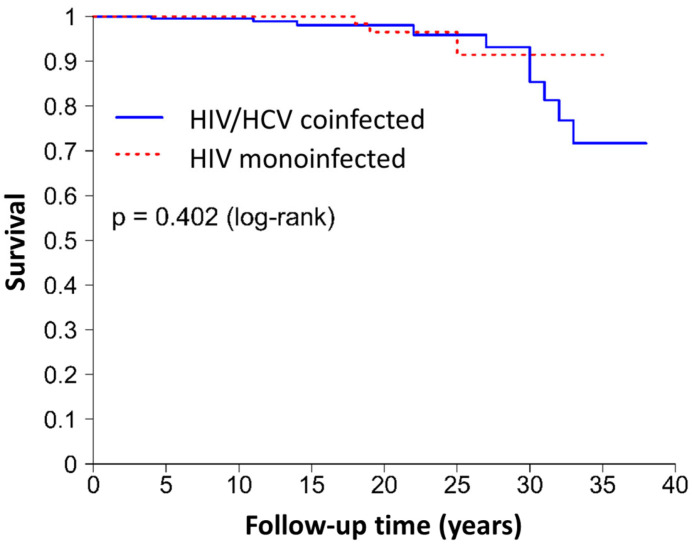
PS analysis results for the risk of comorbidities development and mortality: HIV vs. HIV/HCV cohorts. **Abbreviations.** HCV: Hepatitis C Virus; HIV: Human Immunodeficiency Virus.

**Table 1 jcm-13-03936-t001:** Baseline characteristics of the cohorts.

		HIV/HCV Group (N = 229)	HIV Group (N = 229)	*p*-Value
**Demographics**				
**Age**	Mean (SD)	49.1 (10.8)	49.6 (10.4)	0.627 ^1^
**Gender**				1.000 ^2^
Male	N (%)	218 (95.2%)	218 (95.2%)	
Female	N (%)	11 (4.8%)	10 (4.4%)	
Transgender	N (%)	0 (0.0%)	1 (0.4%)	
**Ethnicity**				0.934 ^2^
African	N (%)	0 (0.0%)	1 (0.4%)	
Arab	N (%)	1 (0.4%)	0 (0.0%)	
Asian	N (%)	1 (0.4%)	1 (0.4%)	
Caucasian	N (%)	174 (76.0%)	176 (76.9%)	
Latino	N (%)	53 (23.1%)	51 (22.3%)	
**Sexual orientation**				<0.001 ^2^
Heterosexual	N (%)	51 (22.3%)	17 (7.4%)	
Homosexual	N (%)	173 (75.5%)	212 (92.6%)	
Bisexual	N (%)	5 (2.2%)	0 (0.0%)	
**Lifestyle habits**				
**Smoking habit**				0.001 ^2^
No	N (%)	97 (42.9%)	122 (53.3%)	
Yes	N (%)	126 (55.8%)	93 (40.6%)	
Former smoker	N (%)	3 (1.3%)	14 (6.1%)	
**Alcohol consumption**				<0.001 ^2^
No	N (%)	120 (53.1%)	167 (72.9%)	
Yes	N (%)	106 (46.9%)	62 (27.1%)	
**Drug consumption (ever)**				<0.001 ^2^
No	N (%)	54 (25.5%)	203 (89.0%)	
Yes	N (%)	157 (74.1%)	25 (11.0%)	
Former consumer	N (%)	1 (0.5%)	0 (0.0%)	
**Purpose of parenteral drugs’ use**				<0.001 ^2^
Nonconsumer	N (%)	117 (55.2%)	226 (98.7%)	
Conventional use	N (%)	48 (22.6%)	2 (0.9%)	
ChemSex	N (%)	47 (22.2%)	1 (0.4%)	
**HIV infection**				
**Time since diagnosis (years)**	Median (IQR)	14.0 (9.0–20.0)	13.0 (9.0–18.0)	0.092 ^3^
**HIV route of transmission**				<0.001 ^2^
Parenteral	N (%)	50 (21.8%)	7 (3.1%)	
Sexual	N (%)	179 (78.2%)	222 (96.9%)	
**HIV clinical stage**				<0.001 ^2^
Stage A	N (%)	140 (61.1%)	163 (71.2%)	
Stage B	N (%)	42 (18.3%)	36 (15.7%)	
Stage C	N (%)	47 (20.5%)	30 (13.1%)	
**CD4+ cell count at ART initiation (cells/µL)**	Median (IQR)	306.0 (180.0–472.0)	314.0 (216.0–440.0)	0.845 ^3^
**CD4+ cell count at ART initiation (range)**				0.012 ^2^
1–199 cells/µL	N (%)	61 (30.0%)	44 (20.2%)	
200–499 cells/µL	N (%)	98 (48.3%)	136 (62.4%)	
≥500 cells/µL	N (%)	44 (21.7%)	38 (17.4%)	
**CD4+/CD8+ ratio at ART initiation**	Median (IQR)	0.28 (0.14–0.47)	0.30 (0.18–0.45)	0.313 ^3^
**Baseline HIV-1 VL (copies/mL)**	Median (IQR)	107.730 (31.960–271.000)	86.743 (23.147–258.250)	0.282 ^3^
**Baseline HIV-1VL (range)**				0.102 ^2^
<100.000 copies/mL	N (%)	93 (48.2%)	114 (54.0%)	
100.000–500.000 copies/mL	N (%)	67 (34.7%)	76 (36.0%)	
<500.000 copies/mL	N (%)	33 (17.1%)	21 (10.0%)	
**History of AIDS**	N (%)	48 (21.0%)	29 (12.7%)	0.025 ^2^
**HIV treatment**				
**Time since HIV diagnosis to treatment (years)**	Median (IQR)	1.0 (0.0–4.0)	1.0 (0.0–3.0)	0.155 ^3^
**Time on HIV treatment (years)**	Median (IQR)	11.0 (7.0–15.0)	11.0 (7.0–15.0)	0.573 ^3^
**Ever exposed to rilpivirine**	N (%)	65 (28.4%)	54 (23.6%)	0.287 ^2^
**Poor previous ART adherence**	N (%)	29 (12.7%)	9 (3.9%)	0.001 ^2^
**HCV infection**				
**HCV route of transmission**				-
Parenteral	N (%)	51 (22.3%)	-	
Sexual	N (%)	178 (77.7%)	-	
**HCV clinical stage at diagnosis**				-
Acute	N (%)	150 (65.5%)	-	
Chronic	N (%)	79 (34.5%)	-	
**Time since HCV diagnosis to successful treatment (years)**	Mean (SD)	5.66 (9.13)	-	
	Median (IQR)	1.0 (0.0–6.0)	-	
**HCV genotypes**				-
1a/b	N (%)	159 (69.5%)	-	
2/3	N (%)	13 (5.6%)	-	
4	N (%)	57 (24.9%)	-	
**HCV viral load at DAA initiation (IU/mL)**				-
<800.000	N (%)	88 (38.4%)	-	
≥800.000	N (%)	141 (61.6%)	-	
**Achieved SVR12**	N (%)	225 (98.2%)		-
**Transient elastography (FibroScan) LS before DAAs (kPa)**	Mean (SD)	7.13 (5.32)	-	-
**Transient elastography (FibroScan) LS before DAAs (Metavir fibrosis score stages)**				-
F0–F1 (<7.1 kPa)	N (%)	153 (66.8%)	-	
F2 (7.1–9.4 kPa)	N (%)	33 (14.4%)	-	
F3 (9.5–12.4 kPa)	N (%)	20 (8.7%)	-	
F4 (≥12.5 kPa)	N (%)	13 (5.7%)	-	
No data	N (%)	10 (4.4%)	-	
**HCV reinfections (number)**				-
2 HCV infections	N (%)	25 (10.9%)	-	
3 HCV infections	N (%)	5 (2.2%)	-	
**Baseline biochemistry results**				
**AST (IU/L)**	Median (IQR)	48.0 (35.0–74.0)	21.0 (17.0–27.0)	<0.001 ^3^
**ALT (IU/L)**	Median (IQR)	69.0 (45.0–123.0)	21.0 (16.0–30.0)	<0.001 ^3^
**GGT (IU/L)**	Median (IQR)	60.5 (32.8–114.0)	21.0 (15.0–32.0)	<0.001 ^3^
**Total cholesterol (mg/dL)**	Mean (SD)	159.0 (33.2)	181.0 (35.5)	<0.001 ^1^
**LDL-cholesterol (mg/dL)**	Mean (SD)	90.4 (27.5)	110.0 (34.1)	<0.001 ^1^
**HDL-cholesterol (mg/dL)**	Mean (SD)	47.3 (13.7)	49.8 (15.2)	0.063 ^1^
**Triglycerides (mg/dL)**	Median (IQR)	101.0 (74.0–138.0)	106.0 (83.0–151.0)	0.136 ^3^
**GFR (mL/min/1.73 m^2^)**	Mean (SD)	94.0 (15.1)	83.0 (17.8)	<0.001 ^3^
**Platelet (x10^3^ µL)**	Mean (SD)	231.0 (66.6)	251.0 (62.7)	0.001 ^1^
**CD4+ (cells/µL)**	Mean (SD)	703.0 (350.)	812.0 (328.0)	0.001 ^1^
**CD8+ (cells/µL)**	Mean (SD)	1074.0 (510.0)	968.0 (442.0)	0.018 ^1^
**CD4+/CD8+**	Mean (SD)	0.73 (0.37)	0.96 (0.49)	<0.001 ^1^
**APRI score**	Median (IQR)	0.55 (0.35–0.90)	0.09 (0.05–0.13)	<0.001 ^3^
**APRI fibrosis stage**				<0.001 ^2^
No fibrosis	N (%)	102 (44.5%)	226 (98.7%)	
Moderate fibrosis	N (%)	94 (41.0%)	3 (1.3%)	
Cirrhosis	N (%)	33 (14.4%)	0 (0.0%)	
**FIB-4 score**	Median (IQR)	1.12 (0.78–1.57)	0.33 (0.22–0.57)	<0.001 ^3^
**FIB-4 fibrosis stage**				<0.001 ^2^
No fibrosis	N (%)	159 (69.4%)	224 (97.8%)	
Moderate fibrosis	N (%)	53 (23.2%)	5 (2.2%)	
Cirrhosis	N (%)	17 (7.4%)	0 (0.0%)	
**Baseline comorbidities**				
**Number of previous comorbidities**	Mean (SD)	0.30 (0.70)	0.10 (0.30)	<0.001 ^1^
**Number of documented STIs**	Mean (SD)	4.20 (3.60)	2.60 (2.40)	<0.001 ^1^
**Hepatitis A test** (Positive)	N (%)	163 (74.4%)	155 (76.4%)	0.730 ^2^
**HBsAg** (Positive)	N (%)	9 (3.9%)	6 (2.7%)	0.662 ^2^
**HBsAb** (Positive)	N (%)	136 (59.4%)	147 (66.8%)	0.125 ^2^
**HBcAb** (Positive)	N (%)	97 (42.4%)	81 (37.0%)	0.287 ^2^
**Obesity (BMI > 30 kg/m^2^)**	N (%)	19 (8.3%)	21 (10.8%)	0.483 ^2^
**Hypertension at diagnosis**	N (%)	18 (7.9%)	3 (1.3%)	0.002 ^2^
**DM at diagnosis**	N (%)	18 (7.9%)	3 (1.3%)	0.040 ^2^
**Dyslipidemia at diagnosis**	N (%)	15 (6.6%)	12 (5.2%)	0.692 ^2^
**CVD at diagnosis**	N (%)	7 (3.1%)	1 (0.4%)	0.075 ^2^
**Kidney disease at diagnosis**	N (%)	6 (2.6%)	0 (0.0%)	0.040 ^2^
**Hepatic disease at diagnosis**	N (%)	15 (6.6%)	0 (0.0%)	<0.001 ^2^
**Non-AIDS cancer at diagnosis**	N (%)	9 (3.9%)	0 (0.0%)	0.007 ^2^

**Abbreviations.** AIDS: Acquired Immunodeficiency Syndrome; ALT: Alanine amino Transferase; ART: Antiretroviral Treatment; AST: Aspartate amino Transferase; BMI: Body Mass Index; CVD: Cardiovascular Disease; GGT: Gamma Glutamyl Transferase; HBcAb: Hepatitis B core antibody; DAA: Direct Acting Antiviral; DM: Diabetes mellitus; Hepatitis B core Antibodies; HBsAb: Hepatitis B surface Antibodies; HBsAg: Hepatitis B surface Antigen; HCV: Hepatitis C Virus; HIV: Human Immunodeficiency Virus; IQR: Interquartile Range; LS: Liver Stiffness; SD: Standard Deviation; STI: Sexually Transmitted Infection; SVR12: Sustained Viral Response 12 months; VL: Viral Load. **Clarifications:** ^1^ Student’s *t* test was used to determine differences between the HIV/HCV and HIV groups. ^2^ Differences between the HIV/HCV and HIV groups according to the chi-square test or Fisher’s exact test. ^3^ Differences between the HIV/HCV and HIV groups according to the Mann—Whitney U test.

**Table 2 jcm-13-03936-t002:** Laboratory outcomes achieved in the HIV/HCV cohort after patients achieved HCV clearance with DAA therapy compared with HIV measurements.

		HIV/HCV Group (N = 229)	HIV Group (N = 229)	*p*-Value
**AST (IU/L)**	Median (IQR)	21.0 (17.0–26.0)	21.0 (17.0–27.0)	0.991 ^1^
**ALT (IU/L)**	Median (IQR)	18.0 (14.0–23.0)	21.0 (16.0–30.0)	<0.001 ^1^
**GGT (IU/L)**	Median (IQR)	18.0 (13.0–31.0)	21.0 (15.0–32.0)	0.059 ^1^
**Total cholesterol (mg/dL)**	Mean (SD)	172.0 (36.2)	181.0 (35.5)	0.010 ^2^
**LDL-cholesterol (mg/dL)**	Mean (SD)	102.0 (33.3)	110.0 (34.1)	0.007 ^2^
**HDL-cholesterol (mg/dL)**	Mean (SD)	46.5 (14.0)	49.8 (15.2)	0.015 ^2^
**Triglycerides (mg/dL)**	Median (IQR)	107.0 (79.0–156.0)	106.0 (83.0–151.0)	0.842 ^1^
**GFR (mL/min/1.73 m^2^)**	Mean (SD)	92.2 (15.2)	83.0 (17.8)	<0.001 ^2^
**Platelet (×** **10^3^ µL)**	Mean (SD)	235.0 (72.6)	251.0 (62.7)	0.011 ^2^
**CD4+ (cells/µL)**	Mean (SD)	752.0 (399.0)	812.0 (328.0)	0.082 ^2^
**CD8+ (cells/µL)**	Mean (SD)	1119.0 (497.0)	968.0 (442.0)	0.001 ^2^
**CD4+/CD8+**	Mean (SD)	0.74 (0.36)	0.96 (0.49)	<0.001 ^2^
**APRI score**	Median (IQR)	0.23 (0.17–0.30)	0.09 (0.05–0.13)	<0.001 ^1^
**APRI fibrosis stage**				0.001 ^3^
No fibrosis	N (%)	208 (90.8%)	226 (98.7%)	
Moderate fibrosis	N (%)	18 (7.9%)	3 (1.3%)	
Cirrhosis	N (%)	3 (1.3%)	0 (0.0%)	
**FIB-4 score**	Median (IQR)	0.94 (0.69–1.29)	0.33 (0.22–0.57)	<0.001 ^1^
**FIB-4 fibrosis stage**				0.001 ^3^
No fibrosis	N (%)	182 (79.5%)	224 (97.8%)	
Moderate fibrosis	N (%)	40 (17.5%)	5 (2.2%)	
Cirrhosis	N (%)	7 (3.1%)	0 (0.0%)	

**Abbreviations.** ALT: Alanine amino Transferase; AST: Aspartate amino Transferase; DAA: Direct Acting Antiviral; GFR: Glomerular Filtration Rate; GGT: Gamma Glutamyl Transferase; HCV: Hepatitis C Virus; HIV: Human Immunodeficiency Virus; IU: International Units; SD: Standard Deviation. **Clarifications:** ^1^ Differences between the HIV/HCV and HIV groups according to the Mann—Whitney U test. ^2^ Student’s *t* test was used to determine differences between the HIV/HCV and HIV groups. ^3^ Differences between the HIV/HCV and HIV groups according to the chi-square test or Fisher’s exact test.

**Table 3 jcm-13-03936-t003:** Development of comorbidities: multivariate logistic regression models.

Variables Included ^1^		OR	CI 95%	*p*-Value
Hypertension	Group (HIV vs. HIV/HCV)	0.94	0.42–2.10	0.878
	Number of documented STIs	0.84	0.73–0.96	0.011
	Drugs consumption (ever)	0.41	0.16–1.07	0.068
	Alcohol consumption	0.89	0.43–1.82	0.742
DM	Group (HIV vs. HIV/HCV)	1.85	0.52–6.63	0.345
	Number of previous comorbidities	3.76	2.03–6.95	<0.001
	Number of documented STIs	0.76	0.58–1.00	0.051
	CD8+ (cells/µL) ^2^	1.10	1.01–1.19	0.024
	HIV route of transmission	1.33	0.30–5.93	0.711
Dyslipidemia	Group (HIV vs. HIV/HCV)	1.06	0.56–1.99	0.857
	Number of documented STIs	0.91	0.83–0.99	0.044
	Drugs consumption (ever)	0.49	0.25–0.97	0.041
CVD	Group (HIV vs. HIV/HCV)	1.60	0.69–3.71	0.269
	Number of previous comorbidities	2.39	1.45–3.96	0.001
Kidney disease	Group (HIV vs. HIV/HCV)	2.50	0.88–7.07	0.084
	Number of documented STIs	0.76	0.60–0.96	0.022
Hepatic disease	Group (HIV vs. HIV/HCV)	0.59	0.05–7.15	0.681
	Time since HIV diagnosis to treatment	1.11	1.02–1.19	0.010
	FIB-4 fibrosis stage (fibrosis vs. no fibrosis)	8.51	1.23–59.00	0.030
Non-AIDS cancer	Group (HIV vs. HIV/HCV)	2.38	0.91–6.21	0.076
	Number of previous comorbidities	1.97	1.09–3.56	0.025
	Number of documented STIs	0.89	0.75–0.172	0.172
Death	Group (HIV vs. HIV/HCV)	0.25	0.04–1.49	0.128
	Number of previous comorbidities	1.49	0.76–2.93	0.240
	Number of documented STIs	0.41	0.21–0.82	0.011
	HIV route of transmission	3.43	0.34–34.35	0.293
	History of AIDS	2.05	0.58–7.22	0.262
	Purpose of parental drugs’ use (No consumption)	1.84	0.19–17.79	0.598

**Abbreviations.** AIDS: Acquired Immune Deficiency Syndrome; CVD: Cardiovascular Disease; CI: Confidence Interval; DAA: Direct Acting Antiviral; DM: Diabetes Mellitus; HCV: Hepatitis C Virus; HIV: Human Immunodeficiency Virus; OR: Odds Ratio; STI: Sexually Transmitted Infection; SVR12: Sustained Viral Response 12 months. **Clarifications:** ^1^ Variables included according to the stepwise method. Patients diagnosed with the comorbidity of interest at baseline were excluded from the specific model. The variable “Group (HIV vs. HIV/HCV)” was included in all models, and a statistically significant OR < 1 indicated that HIV/HCV coinfection was a risk factor for developing the comorbidity. ^2^ For the HIV/HCV cohort, the CD8 T-cell count was calculated after SVR12 with DAA.

**Table 4 jcm-13-03936-t004:** Multivariate logistic analyses results for the HIV and HIV/HCV patient cohorts to measure risk and protective factors.

Comorbidities	Variables Included ^1^	HIV-Monoinfected Patients			HIV/HCV-Coinfected Patients		
		OR	95% CI	*p*-Value	OR	95% CI	*p*-Value
Hypertension	Age	1.09	1.04–1.14	<0.001	1.09	1.03–1.16	0.002
	Time on HIV treatment	1.08	1.01–1.16	0.038	- ^2^	- ^2^	- ^2^
DM	Age	1.10	1.03–1.17	0.005	- ^2^	- ^2^	- ^2^
	Obesity (BMI ≥ 30 kg/m^2^)	6.76	1.77–25.90	0.005	- ^2^	- ^2^	- ^2^
	Number of previous comorbidities	- ^2^	- ^2^	- ^2^	5.50	2.27–8.93	<0.001
	Time on HIV treatment	- ^2^	- ^2^	- ^2^	1.14	1.02–1.28	0.019
Dyslipidemia	Time on HIV treatment	1.13	1.05–1.21	0.001	- ^2^	- ^2^	- ^2^
	Obesity (BMI ≥ 30 kg/m^2^)	7.17	2.42–21.25	<0.001	3.82	1.18–12.37	0.025
	Number of previous comorbidities	0.03	0.01–0.44	0.010	- ^2^	- ^2^	- ^2^
	Age	1.06	1.02–1.11	0.006	- ^2^	- ^2^	- ^2^
	Treatment regimen with rilpivirine	3.20	1.37–7.48	0.007	- ^2^	- ^2^	- ^2^
	CD4+ (cells/µL)	1.14	1.02–1.28	0.019	- ^2^	- ^2^	- ^2^
	Gender (Female vs. Male) ^3^	- ^2^	- ^2^	- ^2^	11.5	2.60–50.60	0.001
	HCV VL (≥800,000 IU/mL vs. <800,000 IU/mL)	- ^2^	- ^2^	- ^2^	3.48	1.29–9.36	0.014
	HIV clinical stage (vs. A)						
	Stage B	- ^2^	- ^2^	- ^2^	2.27	0.88–5.86	0.068
	Stage C	- ^2^	- ^2^	- ^2^	0.38	0.10–1.42	0.174
CVD	HIV follow-up time	1.14	1.06–1.23	0.001	- ^2^	- ^2^	- ^2^
	Number of previous comorbidities	4.87	1.40–16.89	0.013	2.04	1.16–3.58	0.013
	Number of HCV infections	- ^2^	- ^2^	- ^2^	2.57	1.03–6.38	0.042
Kidney disease	Age	1.11	1.05–1.17	<0.001	- ^2^	- ^2^	- ^2^
	Time on HIV treatment	- ^2^	- ^2^	- ^2^	1.15	1.02–1.30	0.027
Hepatic disease	CD4+ (cells/µL)	1.46	1.03–2.05	0.032	- ^2^	- ^2^	- ^2^
	Time since HIV diagnosis	- ^2^	- ^2^	- ^2^	1.17	1.07–1.28	0.001
Non-AIDS cancer	Age	1.16	1.09–1.23	<0.001	- ^2^	- ^2^	- ^2^
	HCV diagnosis phase (chronic vs. acute)	- ^2^	- ^2^	- ^2^	16.20	2.00–138.3	0.009
Death	Age	1.18	1.04–1.33	0.011	- ^2^	- ^2^	- ^2^
	Time since HCV diagnosis to DAA’s treatment	- ^2^	- ^2^	- ^2^	1.13	1.05–1.20	0.001

**Abbreviations.** AIDS: Acquired Immune Deficiency Syndrome; BMI: Body Mass Index; CI: Confidence Interval; CVD: Cardiovascular Disease; DAA: Direct Acting Antiviral; DM: Diabetes Mellitus; HIV: Human Immunodeficiency Virus; OR: Odds Ratio; VL: Viral Load. **Aclarations:** ^1^ Variables included according to the stepwise method. Patients diagnosed with the comorbidity of interest at baseline were excluded from the specific model. ^2^ The lack of data for this variable is due to the fact that the variable was not included in this model according to the stepwise method. ^3^ The category “Transgender” in the variable “Gender” was not included because of the low number of cases.

## Data Availability

The data used for this study may be provided to other investigators upon request of the corresponding author (B.A.A.). The request should include a study protocol and data analysis that will be evaluated by the authors of this study to determine the data extent to which the data will be shared.

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
