# Peer review of "The Era of DAAs: Assessing the Patients’ Characteristics, Clinical Impact, and Emergence of Comorbidities in HIV/HCV-Coinfected versus HIV-Infected Individuals"

_jcm, 2024, doi:10.3390/jcm13133936_

Round 1

Reviewer 1 Report

Comments and Suggestions for Authors

Dear authors, I find interesting the idea behind the article. There is a lot of information mentioned in document difficult to comprehend and process.

1. The design of the study is  difficult to understand. You said you collected data from patients between January 2023 to September 2023. Did these patients come to the hospital, did you make laboratory tests to these patients during the study period? Please add more details about the inclusion and exclusion criteria. Some inclusion criteria are mentioned at the end of the manuscript (line 255): "For the monoinfected group, all patients had a VL<50copies/ml which was an inclusion criterion". The monoinfected HIV group of patients were selected to have no complications and viral load< 50copies/ml, while the HIV/HCV had different viral loads and different complications?  Line 168-173: Complications like hypertension, kidney disease can be also caused by HIV infection. Why did you assume they were only caused by HCV infection?

2. You have a multitude of data regarding the patients, but the title only refers to comorbidities HIV/HCV infection. Either you select the data to present in results section more carefully in accordance with the title, or you change the whole article structure into: patients characteristics on these specific group. 

3. Table 1 and 2 could also contain p values for each category.

4. In Table 4 I find it difficult to understand why there is lack of data. For example: age and obesity in relation with DM for the HIV/HCV group. Why there are only lines and no values?

5. Discussion section is difficult to follow. A wide range of information that is not necessarily structured. Add more comparisons with other literature findings and less assumptions. 

One article should be checked as followed:

- What is the hypothesis? That HCV has an influence in developing comorbidities in HIV/HCV patients

- Materials and methods: "What should I compare to prove that hypothesis?" by selecting two equal cohorts of PLWH, half with HCV co-infection and half without HCV. The two cohorts were matched by age and sex (inclusion criteria should be similar for both cohorts regarding also HIV viral loads and comorbidities - Line 173 - in monoinfected patients, the comorbidities presented at baseline were not considered). You inquire the two cohorts about risk factors (smoking, sex, drugs... ), particularities (age, sex, ethnicity, etc), you take laboratory tests to identify comorbidities (DM, HTA, Dyslipidemia, kidney disease, etc).

-Results: "What are the results of comparing the two cohorts regarding the hypothesis?" In present article there is too much information which is not related to the subject and makes the reader confused about the topic. Please rewrite this section asking for each paragraph: Is this finding important for my hypothesis? 

-Discussion:  "are my results similar to other studies? What did other researchers found about the same subject? How are my findings important for my research and for others?" You compare your presents findings with your previous findings (references 30, 41, 44, 47 are very frequently mentioned). Add more references (different from autocitations) regarding the same subject! and compare your findings with others. 

-Conclusion. "Is my hypothesis in the beginning true or false using the results and discussion?" Conclusion in this article seems fine.

Author Response

Thanks for your comments! We appreciate your suggestions as we believe that these changes greatly improve our manuscript. Next, we provide a point-by-point list of your previous comments and modifications performed to the paper. Additionally, a new version of the text is provided including modifications with the track changes tool to facilitate their identification.

Reviewer comment:

The design of the study is difficult to understand. You said you collected data from patients between January 2023 to September 2023. Did these patients come to the hospital, did you make laboratory tests to these patients during the study period? Please add more details about the inclusion and exclusion criteria. Some inclusion criteria are mentioned at the end of the manuscript (line 255): "For the monoinfected group, all patients had a VL<50copies/ml which was an inclusion criterion". The monoinfected HIV group of patients were selected to have no complications and viral load< 50copies/ml, while the HIV/HCV had different viral loads and different complications?  Line 168-173: Complications like hypertension, kidney disease can be also caused by HIV infection. Why did you assume they were only caused by HCV infection?

Authors response:

Thanks for this suggestion! Regarding the study design and inclusion/exclusion criteria, we provided a deeper insight of both in section “2.1. Study design and participants”. Slight changes were included to avoid being repetitive.

Additionally, your questions were assessed in this section to avoid misunderstanding of potential future readers. The viral load was an exclusion criterion only for coinfected patients as it refers to the HCV viral load. The presence of complications at baseline was not considered as an exclusion criteria as logistic models took them into account.

Reviewer comment:

You have a multitude of data regarding the patients, but the title only refers to comorbidities HIV/HCV infection. Either you select the data to present in results section more carefully in accordance with the title, or you change the whole article structure into: patients characteristics on these specific group. 

Authors response:

Following this suggestion, we modified the study title to “The Era of DAAs: Assessing the Patients’ Characteristics and Clinical Impact and Emergence of Comorbidities in HIV/HCV-Coinfected versus HIV-Infected Individuals” in order to be consistent with the results presented. Thank you for your comment!

Reviewer comment:

Table 1 and 2 could also contain p values for each category.

Authors response:

We have provided p-values for the patients’ characteristics and the inference test conducted for each category as footnote in both tables. The Supplementary Files were also updated with p-Values.

Reviewer comment:

In Table 4 I find it difficult to understand why there is lack of data. For example: age and obesity in relation with DM for the HIV/HCV group. Why there are only lines and no values?

Authors response:

Thanks for your appreciation! The lack of data is explained because the variable is not included in the logistic model according to the stepwise method. We included this clarification as footnote.

Reviewer comment:

Discussion section is difficult to follow. A wide range of information that is not necessarily structured. Add more comparisons with other literature findings and less assumptions.

Authors response:

Thanks for this suggestion! We included changes in the discussion section to address this comment: comparisons vs. previous findings, structured discussion, etc.

We hope that you find the new version of the manuscript adequate for publication. In any case, let us know if there are new issues to be addressed. Thanks again!

Reviewer 2 Report

Comments and Suggestions for Authors

The manuscript provides a comprehensive retrospective analysis of HIV/HCV-coinfected patients and HIV-infected patients and findings are noteworthy, showing that successful HCV elimination using DAAs significantly improves the outlook regarding comorbidities and survival in HIV/HCV-coinfected individuals. The manuscript highlights the importance of continuing efforts toward early detection and treatment initiation, providing an optimistic perspective for those living with HIV/HCV coinfection. Despite some limitations noted by the authors in the final section, particularly concerning sample gathering and following up term, the study remains a valuable resource for researchers and physicians. Additionally, the manuscript is well-organized and clearly written, making it accessible to a broad audience.

Author Response

Thank you very much for comments! In this second round of revision, we provide a new version of the manuscript with several changes that were suggested by other reviewers, which we appreciate as we consider that improve our work. We hope that you still find the manuscript adequate!

Reviewer 3 Report

Comments and Suggestions for Authors

Congratulations to the authors for identifying new aspects of significant scientific relevance in HIV-infected patients. This population, thanks to new therapies that have significantly increased survival rates, now faces a range of comorbidities, primarily cardiovascular, throughout their lives. The analysis conducted on the elimination of HCV infection and its impact on the clinical picture of HIV patients is particularly interesting. As I have already mentioned, the paper is scientifically relevant, well-conducted, and the statistical analysis is satisfactory. I have identified some weaknesses in the paper that, once corrected, can contribute to its improvement:

Authors should give deeper explanations and clinical correlations between clinical outcomes and immunological and biochemical markers;

Authors should justify why did not use the Cox proportional hazards model as this could fully cover all the potential confounder factors that influence mortality and comorbidity;

Moreover in order to enrich your introduction regarding cardiovascular comorbidities in HIV patients please consider:  Catheter ablation approach and outcome in HIV+ patients with recurrent atrial fibrillation. J Cardiovasc Electrophysiol. 2023 Dec;34(12):2527-2534. doi: 10.1111/jce.16076. Epub 2023 Sep 25. PMID: 37746923.

Authors might provide a detailed analysis of how lifestyle factors such as smoking, alcohol and drug use in coinfected group on the progression of  comorbidities and health outcomes.

Comments on the Quality of English Language

just minor english revisions are needed

Author Response

Thanks for revising our manuscript! We appreciate your suggestions as we believe they will improve our work. We attempted to include all your comments in the revised version provided and we present a point-by point list of the changes made.

Reviewer comment:

Authors should give deeper explanations and clinical correlations between clinical outcomes and immunological and biochemical markers.

Authors response:

Thanks for this suggestion, which will help potential readers to understand the clinical implications of immunological and biochemical markers. We have included a deeper insight into this regard in the discussion.

Reviewer comment:

Moreover in order to enrich your introduction regarding cardiovascular comorbidities in HIV patients please consider:  Catheter ablation approach and outcome in HIV+ patients with recurrent atrial fibrillation. J Cardiovasc Electrophysiol. 2023 Dec;34(12):2527-2534. doi: 10.1111/jce.16076. Epub 2023 Sep 25. PMID: 37746923.

Authors response:

Thank you very much for the suggestion and the reference provided. Although you suggested to include it in the introduction, according to the other reviewers comments we have included it in the discussion as suggested by the other reviewers.

Reviewer comment:

Authors might provide a detailed analysis of how lifestyle factors such as smoking, alcohol and drug use in coinfected group on the progression of comorbidities and health outcomes.

Authors response:

Thanks! We have added a paragraph discussing about this regard.

Finally, we would like to mention that we have sent the manuscript to American Journal Experts (AJE), in order to review the English grammar. We can included the certificate provided by the company as a Supplementary File.

We hope that you find the new version of the manuscript adequate for publication. In any case, let us know if there are new issues to be addressed. Thanks again for your time spent and dedication!

Round 2

Reviewer 1 Report

Comments and Suggestions for Authors

I have seen the improvements you`ve made and they look satisfactory. You uploaded the English-proof certification of another article.

You have added p-values in table 1 and Table 2, now please interpret them in text and then compare them in Discussion section.

Other older comments of mine seem resolved.

Author Response

Thanks for your comments! First we have updated the title of the Certificate to fit with the new title of the manuscript. Moreover, following the reviewer’s suggestion, we have provided a deeper insight of the interpretation of the relevant findings derived from the estimated p-values in the “Results” section (now presenting p-values whereas previous version described tendency in results as p-values were not presented). Although the “Discussion” section already assess these differences, additional interpretation of the observed differences was included.

Thanks again for your time and dedication! We believe that taking into consideration your comments has improved notable the manuscript.

Reviewer 3 Report

Comments and Suggestions for Authors

congratulations to the authors for the improved version of the manuscript

Author Response

Thanks again for your time and dedication! We believe that your suggestions were key to improve our work.